# Cytotoxic Withanolides from the Whole Herb of *Physalis angulata* L.

**DOI:** 10.3390/molecules24081608

**Published:** 2019-04-23

**Authors:** Qinghong Meng, Jiajia Fan, Zhiguo Liu, Xiwen Li, Fangbo Zhang, Yanlin Zhang, Yi Sun, Li Li, Xia Liu, Erbing Hua

**Affiliations:** 1College of Biotechnology, Tianjin University of Science & Technology, Tianjin 300457, China; qinghongmeng0813@hotmail.com (Q.M.); hxu@icmm.ac.cn (Y.Z.); 2Institute of Chinese Materia Medica, China Academy of Chinese Medical Sciences, Beijing 100010, China; Jiajia.Fan@pfizer.com (J.F.); liuzhiguo1321@163.com (Z.L.); xwli@icmm.ac.cn (X.L.); fbzhang@icmm.ac.cn (F.Z.); 3Institute of Materia Medica, Chinese Academy of Medical Sciences & Peking Union Medical College, Beijing 100010, China; 4School of Chemistry, Chemical Engineering and Life Science, Wuhan University of Technology, Wuhan 430070, China; lrx1125@126.com

**Keywords:** Solanaceae, *Physalis angulata* L., withanolides, cytotoxic activities

## Abstract

*Physalis angulata* L. is a medicinal plant of the Solanaceae family, which is used to produce a variety of steroids. The present study reports on the cytotoxic withanolides of this plant. The species of *Physalis angulata* L. was identified by DNA barcoding techniques. Two new withanolides (**1**–**2**), together with six known analogues (**3**–**8**)**,** were isolated from the whole plant of *Physalis angulata* L. The structures of these new compounds were determined on the basis of extensive spectroscopic data analyses and electronic circular dichroism (ECD) calculations. The withanolides exhibited strong cytotoxic activities against A549, Hela and p388 cell lines. Furthermore, compounds **1** and **2** induced typical apoptotic cell death in A549 cell line according to the evaluation of the apoptosis-inducing activity by flow cytometric analysis.

## 1. Introduction

*Physalis angulata* L. belongs to the family of Solanaceae, which is frequently used as traditional medicine in China. It generally grows in valleys or country roadsides at an altitude of 500–5000 m [1]. The whole herb of *Physalis angulata* L. has been widely used to relieve inflammatory conditions, diabetes, anemia and cancer [2]. Its main constituents include withanolides, terpenoids, carotenoids, flavonoids and polysaccharides [3]. The medicinal plants of the genus *Physalis* are known to produce withanolides, which structurally have an ergostane skeleton [4]. Most withanolides are polyoxygenated and their structures can be divided into two types of δ-lactone/lactol and γ-lactone/lactol based on the differences between their substituted groups at C-17 side chain. Most withanolides isolated from the genus *Physalis* belong to the δ-lactone/lactol type, which have different modified skeletons, such as physalins, neophysalins and withaphysalins. These modified withanolides have the characteristic groups of 5β,6β-epoxides, 5-ene withanolides and 6α,7α-epoxides [1]. More than 100 withanolides containing the modified skeletons were isolated from *P. alkengi*, *P. pubescens* and *P. angulata*. [5,6,7]. Withanolides have multiple pharmacological effects, such as anti-tumor, anti-stress, immunosuppressive, anti-microbial and anti-inflammatory activities. Many significant pharmacological studies focusing on the active withanolides from *Physalis* species have been reported, such as *P. alkekengi* [5], *P. angulata* [8], *P. minima* [9] and *P. pubescens* [10]. In our study that focused on the discovery of antitumor agents from the genus *Physalis*, we investigated the medicinal plant of *Physalis angulata*. Previous studies on this plant demonstrated that its CH_2_Cl_2_ extract possessed cytotoxic activity against A549 (human non-small cell lung cancer cell lines; IC_50_ value: 22.4 μg/mL). Further bioactivity-guided fractionation of the CH_2_Cl_2_-soluble layer led to the isolation and identification of two new withanolides (1–2) and six known analogues of physagulin F (3) [11], physagulin K (4) [12], Physalin B (5) [13], Physalin F (6) [14], Physalin H (7) [15] and 5α-ethoxy-6β-hydroxy-5,6-dihydrophysalin B (8) [16] (Figure 1). Compounds **1**–**8** have different modified skeletons of δ-lactone/lactol type, among which **1**–**4** belong to physalin and **5**–**8** belong to withaphysalins. Both types are common and widely found in the genus *Physalis*. The cytotoxic activities for all the withanolides against A549, Hela (human cervical cancer cell lines) and p388 (human leukemia cell lines) cell lines were evaluated using the MTT assay. Further studies showed that **1** and **2** induced cell apoptosis in A549 cell lines in a dose-dependent manner. The known compounds were identified by the comparison of spectroscopic data with those reported. 

## 2. Results and Discussion

### 2.1. Structure Elucidation of Compounds ***1***–***2***

The molecular formula of compound **1** was determined to be C_28_H_34_O_9_ by HR-ESIMS. The analysis of the ^1^H-NMR data in conjunction with the HSQC spectrum revealed the presence of two oxygenated methines, an oxygenated methylene, three olefinic protons, six alphatic methylenes, three alphatic methines and three singlet methyls (Table 1, Appendix A). The whole characteristic data of ^1^H- and ^13^C-NMR indicated that **1** was a 13, 14-seco-withanolide analogue of minisecolide C [17]. The olefinic proton signals at δ_H_ 5.65 (1H, dt, *J* = 9.4, 4.0 Hz), 6.08 (1H, d, *J* = 9.4 Hz) and 5.69 (1H, dd, *J* = 5.2, 2.2 Hz) in the ^1^H-NMR, combined with the HMBC correlations from H-4 to C-2, C-3, C-6 and C-10 as well as from H-6 to C-5 and C-10, indicated that conjugated double bonds existed between C-3 and C-6. The comparison of the ^13^C-NMR data of **1** with minisecolide C implied differences in rings C and D. The HMBC correlations (Figure 2) from H-8 to C-14 as well as from H-15 to C-13, C-14 and C-16 suggested the presence of a hemiketal group at C-14 and a hydroxyl group at C-15, which also confirmed the presence of an epoxy bridge at C-13 and C-14. Further evidence in the HMBC correlations from H-28 to C-23, C-24 and C-25 indicated a primary hydroxyl group at C-28 instead of a methyl group. The relative configuration of **1** was determined by the NOESY spectrum (Figure 2). The NOESY correlations of H-15/H-17 suggested a β-orientation of OH-15. The (*R*)-configuration at C-22 was confirmed by the positive Cotton effect at 250 nm in the ECD spectrum (Figure 3). The ECD data elucidated that the boat conformation for ring A was more stable when it was in the tetracyclic structures. Thus, the structure of **1** was defined as (22R)-13,14-epoxy-14,15,28-trihydroxy-1-oxo-13,14-secowitha-3,5,24-trien-18,20:22,26-diolide.

Compound **2** was obtained as an amorphous solid. The molecular formula (C_28_H_41_O_6_) was determined by HR-ESIMS *m*/*z* 473.28772 [M + H]^+^ and NMR data (Table 1, Appendix A). The ^1^H-, ^13^C-NMR and HSQC spectra of **2** revealed the presence of four methyls [δ_H_ 1.02 (3H,s), δ_C_ 12.84; δ_H_ 1.24 (3H,s), δ_C_ 20.83; δ_H_ 1.75 (3H,s), δ_C_ 12.15 and δ_H_ 1.91 (3H, s), δ_C_ 20.06], an oxygenated methylene [δ_H_ 3.32 (1H,m),3.43 (1H, dd, *J* = 11.8, 6.6 Hz, δ_C_ 56.96], two oxygenated methines [δ_H_ 3.15 (1H, br s), δ_C_ 59.40 and δ_H_ 4.10 (1H, dd, *J* = 13.2, 3.5 Hz), δ_C_ 80.21] and eight quaternary carbons [including two carbonyls (δ_C_ 212.57 and 165.76) and two olefins (δ_C_ 149.99 and 120.18)]. The above-described data suggested that **2** possessed a characteristic skeleton of withanolide (Figure 1), which was similar to withanolide D [18] and physagulide P [19]. The HMBC correlations (Figure 2) from H-6 (δ_H_ 3.15) to C-5 (δ_C_ 63.76) and C-7 (δ_C_ 31.32) as well as from H_3_-19 (δ_H_ 1.02) to C-5 and C-10 (δ_C_ 51.63) revealed the presence of a 5β, 6β-epoxide moiety at C-5 and C-6 [20]. The ^1^H-^1^H COSY cross-peaks of H-2/H-3/H-4 and H-6/H-7/H-8/H-9 also showed the positions of the epoxide group. Moreover, an α, β-unsaturated-δ-lactone system with α, β-dimethyl groups was elucidated through the HMBC correlations of H-22 (δ_H_ 4.10)/C-24 (δ_C_ 149.99), H-22/C-26 (δ_C_ 165.76), H_2_-23(δ_H_ 2.33 and 2.22)/C-25(δ_C_ 120.2), H_3_-27(δ_H_ 1.75)/C-26 and H_3_-28(δ_H_ 1.91)/C-25. The differences observed between **2** and withanolide D were related to the signals on ring A as the olefin and hydroxyl groups observed in withanolide D were absent in **2**. However, three methylenes [δ_H_ 2.23, 2.71; δ_C_ 34.45; δ_H_ 1.80; δ_C_ 17.55 and δ_H_ 1.15, 1.90; δ_C_ 29.52] replaced these positions in **2**, which was confirmed subsequently by HMBC correlations from H_2_-2 to C-4, from H_2_-3 to C-1 as well as from H-4 to C-5 and C-6. The β-orientation of the 5,6-epoxide was deduced by the NOESY correlations between H-6 and H-4α (δ_H_ 1.90). The experimental ECD curve of **2** was similar to the computed curves of **2**a (Figure 3). A negative Cotton effect was found at 294.5 nm, which confirmed the cis-linkage of A and B rings and the boat conformation of A ring in **2**. The absolute configurations shown in **2** were used as the input configuration of theoretical calculations [21]. By comparing the calculated and experimental ECD spectra, the configuration of **2** was established. Therefore, compound **2** was identified as (17*S*,20*R*,22*R*)-5β,6β-epoxy-18,20-dihydroxy-1-oxowitha-24-enolide.

### 2.2. Cytotoxicities and Cell Apoptosis

All withanolides were evaluated for their cytotoxicity against p388, Hela and A549 human carcinoma cells using the MTT assay. Most compounds exhibited strong cytotoxicity against the tumor cell lines [15,22] (Table 2). Compounds **1** and **2** were further evaluated for their effect on cell apoptosis in A549 cells at three different concentrations for 24 h. The results were analyzed by flow cytometry and demonstrated that the total apoptotic cell population induced by 49.2% at 10.0 μM for **1** while this was 46.6% at 20.0 μM for **2**. This implied that the two compounds induced A549 cell apoptosis in a dose-dependent manner (Figure 4).

### 2.3. Identification of Physalis Angulata L. by DNA Barcoding Techniques

The identification of the plant was carried out via DNA barcoding techniques [23]. The sample included the whole plant that was dried, which was collected in Hainan province. The PCR results indicated that the ITS2 regions were amplified by the universal primers ITS2F/ITS3R. Based on the ITS2 online database, the ITS2 sequences (with a length of 215 bp) could successfully identify a plant sample at the genus level. The results confirmed that the plant sample had an identity that was 99% similar to *Physalis angulata* L.

## 3. Materials and Methods

### 3.1. General Experimental Procedures

The optical rotations were measured with a Perkin-Elmer 241 polarimeter. The UV spectra were conducted on a Shimadzu UV-2201 spectrometer. ECD spectra were recorded on a JASCO J-815 spectrometer. NMR spectra were recorded on a Bruker ARX-600 spectrometer (600 MHz, Bruker Co., Ltd, Karlsruhe, Germany). LC-MS data were obtained using LTQ Orbitrap velospro (Thermo Fisher Scientific Co., Ltd., Bremen, Germany). High performance liquid chromatography (HPLC) was performed on Agilent 1260 pump (Agilent Technologies Co., Ltd., Palo Alto, CA, USA), coupled with Agilent analytical, semi-preparative or preparative Kromasil Eternity XT-5-C18 columns (250 mm × 4.6 mm and 250 mm × 10 mm, respectively), Kromasil Eternity XT-5-PhenylHexyl columns (250 mm × 4.6 mm and 250 mm × 10 mm, respectively) and Cosmosil 5C18-MSII (250 mm × 4.6 mm and 250 mm × 20 mm, respectively). TLC and preparative TLC were monitored on precoated silica gel plates (GF254, Qingdao Haiyang Chemical Co., Ltd., Qingdao, China). Column chromatography was performed on Silica gel (200−300 mesh, Qingdao Marine Chemical Ltd., Qingdao, China) and Sephadex LH-20 (Pharmacia Fine Chemical Co., Ltd., Uppsala, Sweden). 

### 3.2. Extraction and Isolation

The air-dried entire plant of *P. angulata* L. (6.2 kg) was crushed and extracted with ethanol/water under reflux (95:5, 3 times, for 2 h). The solvent was evaporated under a vacuum and the resulting extract was partitioned by CH_2_Cl_2_, EtOAc and *n*-BuOH, respectively. After this, the concentrated layer of CH_2_Cl_2_ (194 g) was chromatographed on a silica gel column using a gradient of CH_2_Cl_2_-MeOH (100:1 to 0:100), which yielded seven fractions (Fr.1–Fr.7). Fr.2 was chromatographed on silica gel using CH_2_Cl_2_-MeOH (25:1, 15:1, 12:1, 8:1, 5:1, 2:1 ) to give 4 fractions (Fr.2.1–Fr.2.4), among which Fr.2.1 was further isolated by silica gel column using CH_2_Cl_2_-acetone (40:1) to yield **5** (50 mg) and Fr.2.2 was isolated over ODS using MeOH-H_2_O (50:50) to create 3 subfractions (Fr.2.2.1–Fr.2.2.3). After this, Fr.2.2.1 was purified by HPLC (methanol/water, 33–50% methanol in 70 min) to yield compounds **1** (9 mg) and **2** (6 mg). Fr.2.2.2 was isolated by HPLC (acetonitrile/water, 30–50% acetonitrile in 65 min), creating compounds **6** (6 mg), **7** (3 mg) and **8**. Finally, Fr.2.4 was purified by HPLC (Methanol/water, 35–55% Methanol in 70 min) to yield **3** (7 mg) and **4** (12 mg).

### 3.3. Characterization of New Compounds ***1*** and ***2***

Compound **1**: white amorphous solid; [α${]}_{D}^{25}$ +21.3 (*c* 0.10, MeOH); HR-ESIMS (positive ions) of **1**: [M + H]^+^ at *m/z* 515.2251 (calcd for C_28_H_35_O_9_, 515.2254). NMR: ^1^H, ^13^C, ^1^H-^1^H COSY, HSQC, HMBC and NOESY data, see Table 1 and Figure 2. CD spectrum, see Figure 3.

Compound **2:** white amorphous solid; [α${]}_{D}^{25}$ +11.5 (*c* 0.10, MeOH); CD (MeOH) 248 (Δε +3.0) nm, 294.5 (Δε −14.7) nm; HR-ESIMS (positive ions) of **2**: [M + H]^+^ at *m/z* 473.2877 (calcd for C_28_H_41_O_6_, 473.2874). NMR: ^1^H, ^13^C, ^1^H-^1^H COSY, HSQC, HMBC and NOESY data, see Table 1 and Figure 2. CD spectrum, see Figure 3.

### 3.4. ECD Calculations

First, a systematic conformational analysis was performed to find all possible conformers within a 3 kcal/mol energy window in the MMFF94 force field. The obtained conformers were further optimized and identified as the stable conformers at the B3LYP/6-31G (d) level by the Gaussian 09 program. The main conformers (Boltzmann distribution > 1%) of compound 1 were chosen for ECD calculations. All quantum computations are performed on an IBM cluster machine located at the High Performance Computing Center of Peking Union Medical College. The lowest 120 electronic excitations were calculated and the energies, oscillator strengths and rotational strengths (velocity) of each electronic excitation were subsequently obtained. After this, ECD spectra were simulated with a half-bandwidth of 0.40 eV.

### 3.5. Cell Culture

The human carcinoma cell lines (Hela, p388 and A549) were obtained from National infrastructure of cell line resource. The cells were maintained in DMEM containing 10% Fetal Bovine Serum (FBS) and 0.4% Penicillin–Streptomycin Solution (100×) at 37 °C under 5% CO_2_.

### 3.6. MTT Assay for Cytotoxicities

The cytotoxic activities of **1**–**8** against Hela, p388 and A549 cells lines were determined using the 3-(4,5-dimethyl-2-thiazolyl)-2, 5-diphenyl-2-H-tetrazolium bromide (MTT) assay. When the cells showed logarithmic growth, they were diluted to a concentration of 1 × 10^4^ cells /mL. The diluted cell suspensions (200 μL) were added into 96-well microtiter plates and incubated at 37 °C under an atmosphere of 5% CO_2_ for 24 h. The test medium dissolved in DMSO was added to each well containing tumor cell medium. After this, the cultures were incubated at 37 °C for 72 h. After the addition of 50 μL of the MTT solution (5 mg/mL) to each well, the plate was incubated for 4 h under the same conditions to stain live cells. The supernatants were removed and the cells were dissolved in 150 μL of DMSO to determine the IC_50_ values. The absorption was measured at 570 nm. Adriamycin and DMSO were used as the positive control and negative control, respectively. The IC_50_ values and standard deviations (±) were determined using Microsoft Excel software from dose–response curves obtained from at least three independent experiments.

### 3.7. Cell Apoptosis Assay

A FITC annexin V apoptosis detection kit (BD) was applied to detect the cell population in the early and late apoptosis stages. The A549 cells were seeded in six-well plates and allowed to grow overnight. These were treated with or without **1** (5.0, 10.0, 20.0 μM) and **2** (10.0, 20.0, 40.0 μM) for a 24 h time period, stained with annexin V-FITC and PI solution before being examined and analyzed quantitatively by a flow cytometer (Becton-Dickinson, San Jose, CA, USA). Adriamycin (2.34 μM) was used as a positive control. Dose–response results were determined based on the average values of three parallel experiments.

### 3.8. Plant Material and Identification by DNA Barcoding Techniques

The entire plant of *P. angulata* L. was collected in Hainan Province, China and authenticated by the deputy researcher Zheng Xi-long, Institute of Chinese Materia Medica, China Academy of Chinese Medical Sciences. The whole plant included its flowers and fruits. A voucher specimen (voucher code: zys2013091) was deposited in the Institute of Chinese Materia Medica, China Academy of Chinese Medical Sciences.

A total of 30 mg of each dry sample was turned in a powder form in a tissue grinder that ran for 2 min at 50 Hz (Sceintz-48, China). The total genomic DNA was isolated using the Plant Genomic DNA Kit (DP305, Tiangen Biotech Co., China). The ITS2 region was amplified using the published universal primers ITS2F (5′-ATGCGATACTTGGTGTGAAT-3′) and ITS3R (5′-GACGCTTCTCCAGACTACAAT-3′) (Sangon Co.), China. This also included 25 μL of a PCR reaction volume containing < 100 ng of genomic DNA, 2× Taq PCR MasterMix (Aidlab Biotechnologies Co., Beijing, China), 1 μL of each primer (2.5 μM) and distilled deionized water. PCR (Eppendorf AG, Hamburg, Germany) conditions were: 1 cycle of 94 °C for 5 min; 35 cycles of 94 °C for 60 s; 55 °C for 60 s and 72 °C for 90 s; followed by 1 cycle of 72 °C for 7 min. PCR products (6 μL of each) were examined on 1.2% agarose gel electrophoresis in 1× TAE buffer for 20 min at 120 V. The purified PCR products were sequenced bidirectionally with PCR primers on an ABI-3730 sequencer (Applied Biosystem, USA). 

Sequences were assembled using the Codoncode aligner v. 4.2.4. The complete ITS2 sequences were annotated based on the Hidden Markov Model (HMM). All ITS2 sequences were deposited in GenBank. Sequence alignment was performed by ClustalW in MEGA5.1. DNA barcoding database for herbal materials (http://www.tcmbarcode.cn/en/) and nucleotide database in National Center for Biotechnology Information (NCBI) (http://blast.ncbi.nlm.nih.gov/Blast.cgi) were used as sources for complementary ITS2 data for species identification.

## 4. Conclusions

Two novel cytotoxic withanolides **1**–**2**, together with six known analogues **3**–**8,** were isolated from the whole herb of *P. angulata* L. The structures of the new compounds were identified by NMR, HR-ESIMS and ECD spectra data. The cytotoxic activity study revealed that compounds **1**–**8** exhibited cytotoxicity against A549, p388 and HeLa cell lines, among which **5**–**8** with a physalin skeleton showed stronger activities than the withaphysalin type of **1**–**4**. Compounds **1** and **2** induced cell apoptosis in A549 cell lines in a dose-dependent manner.

## Figures and Tables

**Figure 1 molecules-24-01608-f001:**
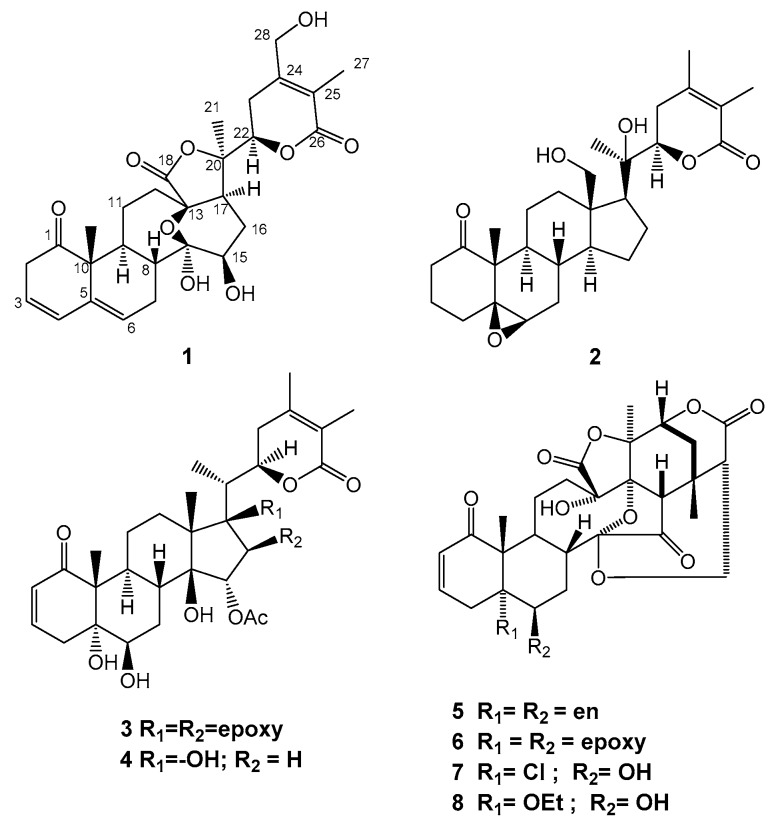
Structures of compounds **1**–**8** from *P. angulata* L.

**Figure 2 molecules-24-01608-f002:**
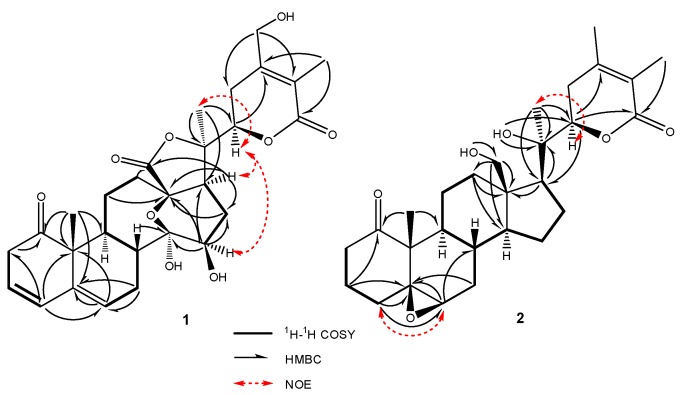
Key HMBC, NOE and ^1^H-^1^H COSY correlations of **1** and **2**.

**Figure 3 molecules-24-01608-f003:**
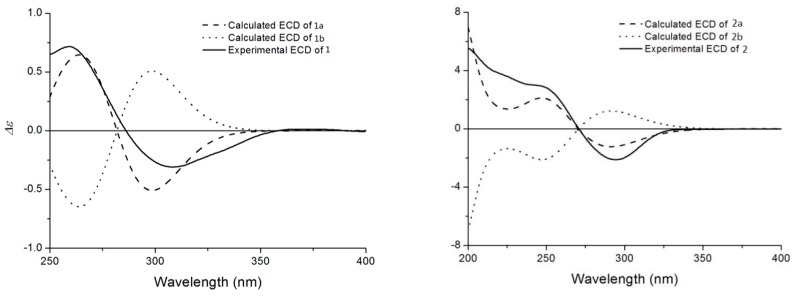
Calculated and experimental ECD spectra for **1** and **2** in MeOH.

**Figure 4 molecules-24-01608-f004:**
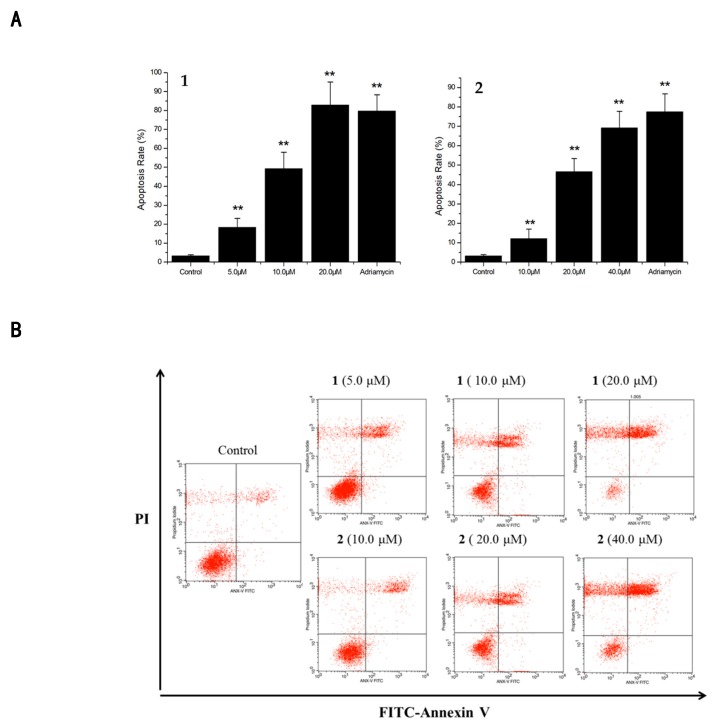
A549 cell apoptosis induced by **1** and **2**. (**A**) Annexin V/FITC flow cytometry to detect the effect of different concentrations of **1** (5, 10, 20 μM) and **2** (10, 20, 40 μM) on apoptosis rate. *n* = 3, ** *p* < 0.01. (**B**) Apoptotic rates of A549 cells as induced by compounds **1** and **2,** which were inspected by Annexin V-PI. Controls were normal A549 cells; A549 cells treated with various concentrations of compounds **1** and **2** for 24 h. Each experiment was repeated 3 times. Data are represented as mean ± SD, *n* = 3, ***p* < 0.01 vs. Control.

**Table 1 molecules-24-01608-t001:** NMR data of **1** and **2** in DMSO-*d*_6_ (600 MHz for ^1^H-NMR and 150 MHz for ^13^C-NMR).

Position	1		2	
	δ_H_ (*J*, Hz)	δ_C_ (ppm)	δ_H_ (*J*, Hz)	δ_C_ (ppm)
1		210.2		212.5
2	3.40 (d, 20.5) 2.62 (dd, 20.5, 4.0)	40.4	2.22 (m) 2.71 (dt, 14.2, 8.3)	34.5
3	5.65 (dt, 9.4,4.0)	122.0	1.80 (m)	17.6
4	6.08 (d, 9.4)	128.3	1.15 (m), 1.90 (m)	29.5
5		139.3		63.76
6	5.69(dd, 5.2, 2.2)	127.3	3.15 (br s)	59.40
7	2.27(m), 2.02(m)	26.4	1.33 (m), 2.00 (dd, 11.0, 2.0)	31.32
8	1.81(m)	44.5	1.31(m)	28.57
9	2.04(m)	32.9	1.16 (m)	41.94
10		52.1		51.63
11	2.48 (m), 1.77 (m)	20.1	1.58 (m), 1.78 (m)	20.96
12	2.38 (m), 1.83 (m)	38.2	0.91 (td, 13.0, 3.5), 2.31(m)	33.43
13		81.0		46.66
14		98.6	1.09 (m)	54.50
15	4.19 (dd, 7.7, 4.6)	74.3	1.08 (m), 1.58 (m)	23.27
16	1.85 (m), 1.41 (m)	24.0	1.58 (m), 1.80 (m)	21.00
17	2.46 (m)	57.7	1.43(t, 9.2)	55.22
18		175.3	3.32 (m), 3.43 (dd, 11.8, 6.6)	56.96
19	1. 31 (s) 0	20.5	1.02 (s)	12.84
20		84.7		73.82
21	1.46 (s)	25.5	1.24 (s)	20.83
22	4.66 (dd, 12.6, 3.4)	76.5	4.10 (dd, 13.2, 3.5)	80.21
23	2.64 (m), 2.40 (m)	26.0	2.22 (m), 2.33 (m)	30.98
24		152.6		149.99
25		119.4		120.18
26		164.6		165.76
27	1.73 (s)	11.5	1.75 (s)	12.15
28	4.21 (d, 14.8) 4.16 (d, 14.8)	60.0	1.91 (s)	20.06
OH-18			5.12 (dd, 6.6, 4.7)	
OH-20			5.36 (s)	

**Table 2 molecules-24-01608-t002:** IC_50_ values of withanolides **1**–**8** against human tumor cell lines.

Cell Lines	IC_50_ (μM) Values of Compounds								
	1	2	3	4	5	6	7	8	Adriamycin
**P388**	8.03 ± 0.02	17.60 ± 0.11	>30	19.21	7.85 ± 0.13	12.50 ± 0.22	5.50 ± 0.12	13.30 ± 0.77	3.20 ± 0.01
**Hela**	21.75 ± 0.23	>30	>30	>30	3.55 ± 0.15	8.50 ± 0.16	2.82 ± 0.03	10.51 ± 0.51	1.50 ± 0.02
**A549**	11.36 ± 0.26	23.51 ± 0.17	17.73 ± 0.39	22.20 ± 0.75	2.10 ± 0.09	15.98 ± 0.13	1.91 ± 0.07	7.62 ± 0.02	2.30 ± 0.10

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
