# Peer review of "Cytotoxic Withanolides from the Whole Herb of Physalis angulata L."

_molecules, 2019, doi:10.3390/molecules24081608_

Round 1

Reviewer 1 Report

Ref.: Molecules-469847

Title:  Cytotoxic withanolides from the whole herb of Physalis angulata L.

Authors: Meng et al.

Reviewer’s comments for author(s)

From my point of view, the manuscript should not be accepted in its present form.

The study should be enlarged to compare the chemical composition from Physalis species.

The used of the DNA barcoding technique is not properly justified. The plant is not unknown, and it was identified by an expert and a voucher was deposited (although no voucher number is referred).

The authors do not refer the plant stage. Vegetative or flowering?

The authors performed just one analysis on one plant sample, and this is no longer acceptable from the biological point of view. The experience should be repeated in time, with other batches of material to confirm the presence of these compounds and if the level at which they exist can be potentially interesting.

From the chemical group classification point of view, which is the difference between “withanolides, terpenoids, carotenoids”. Aren´t they all terpenes?

Please check the document for fonts that give incorrect symbols (squares).

The present study can be used as the basis for a paper reporting repeated analyses of a particular species, and with other species of the same genus with the aim of describing its natural variation.

There are other recent studies on the same topic and the authors do not provide a comparison with previous data.

From the editorial point of view the manuscript also deserves some English revision.

Author Response

Dear reviewer,

Thanks for your comments and suggestions. We have revised the paper according to your advices, and made some changes to English. Please see the responses as follows.

(1) The study should be enlarged to compare the chemical composition from Physalis species.

Response: In accordance with your suggestion, we have added the description as follows: “Most withanolides isolated from the genus Physalis belong to the d-lactone/lactol type, which have different modified skeletons, such as physalins, neophysalins, and withaphysalins. These modified withanolides have the characteristic groups of 5b,6b-epoxides, 5-ene withanolides, and 6a,7a-epoxides. More than 100 withanolides containing the modified skeletons were isolated from P. alkengi, P. pubescens, and P. angulata.”

(2) The used of the DNA barcoding technique is not properly justified. The plant is not unknown, and it was identified by an expert and a voucher was deposited (although no voucher number is referred).

Response: Thanks for your comment. As you noticed, the medicinal plant was identified as Physalis angulata according to its morphological identification by the expert. Because the morphological characteristics of the 5 species (Physalis genus) in China were very similar, we also used the DNA barcoding technique to discriminate and confirm its species. The DNA barcoding technology for identifying TCM has been recorded in Chinese Pharmacopoeia. As the identification of DNA barcoding is not very important section in the manuscript, we have changed the “2.1 Identification of Physalis angulata L. by DNA barcoding techniques” to the last section of 2.3 in Results and Discussion” based on your comment. Moreover, the experiment of DNA barcoding has been removed to the last section. We hope that you can understand our thoughts on the experiment.

(3) The authors do not refer the plant stage. Vegetative or flowering ?

Response: We have replied to it according to the suggestion (Line 154, page 7). The whole plant included its flowers and fruits.

(4) The authors performed just one analysis on one plant sample, and this is no longer acceptable from the biological point of view. The experience should be repeated in time, with other batches of material to confirm the presence of these compounds and if the level at which they exist can be potentially interesting.

Response: Thank you for your comment. We agree that the work is more intuitive to analyze the components and the constituents in batches than to isolate compounds in a single species. The chemical analysis needs to repeat in times. But the advantage of phytochemical work is that it can clarify the bioactive fractions and compounds in the plant. For example, if we would like to study the bioactive constituents and to evaluate the bioactivities for the compounds, we have to isolate the active compounds from the plant by the chromatography methods. This work needs to collect enough amount of the plant for the isolation of active compounds. There are some published papers that reported the bioactive wthanolides from Physalis angulata in the journals of natural products. We accept your suggestion. Thanks!

(5) From the chemical group classification point of view, which is the difference between “withanolides, terpenoids, carotenoids”. Aren´t they all terpenes?

Response: Carotenoids are terpenes, but withanolides belong to the steroids, which have an ergostane skeleton.

(6) Please check the document for fonts that give incorrect symbols (squares).

Response: We have revised them in Line 224 in the manuscript.

(7) The present study can be used as the basis for a paper reporting repeated analyses of a particular species, and with other species of the same genus with the aim of describing its natural variation.

There are other recent studies on the same topic and the authors do not provide a comparison with previous data.

Response: Thank you for your suggestion. We are also studying the other species in the genus Physalis with the aim of finding more active compounds. We have supplemented four more papers as the references to support our manuscript based on your comment, and also compared the data of the known compounds with the reported literatures of [12-17]. We wrote “The known compounds were identified by comparison of spectroscopic data with those reported. “ (Line54). Please check the references of 12-17 (Line47-49) in the manuscript.

Thanks!

Best wishes,

Yi Sun

Reviewer 2 Report

 This manuscript is very well written but I have some comments. First, the withanolides are known as cytotoxic agents, and the new obtained of the authors compounds do not indicate better cytotoxic effect as reference adriamycin. Secondly, the yield of the important compounds are very low and I am not sure if this compounds have perspective as drugs.

Author Response

Dear reviewer,

Thanks for your comments and suggestions. We have revised the paper according to your advices. Please see the responses as follows.

The present manuscript report two news withanolides in Physalis angulata, describing the structures and cyototxic activities in cancer cell lines.

(1) Latests publication related to Physalis angulata are not included in the manuscript (Gao et al 2018; etc.)

Response: Thank you for your comment. We have referred 4 additional papers including references 9, 10, 11, (Line 34-39, page 2) and 21 (Line 87, page 3) in the text.

(2) Table 2 are not mentioned in the text. It is possible to mention it in lines 113-114. 

Response: We added “Table 2” in the text for explaining the results of cytotoxicities (Line 114, page 6).

(3) It is necessary you mention what kind of cancer cell lines are used in the research, by other hand it would be recommendable to use a non-tumoral cell line to know a possible selective effect of compounds studied.

Response: We have explained the types of three cancer cells in the section of “Introduction”.

(4) It is necessary to indicate the concentration of adriamycin used in the experiments of apoptosis MTT.

Response: We have added the concentration of Adriamycin (2.34 mM) in the experiment of apoptosis in the manuscript.

(5) Finally, some comments related to the form of the manuscript:-In line 216 the letter type is changed.

Response: We have revised the letter as “mM” in that part.

Thanks!

Best wishes,

Yi Sun

Reviewer 3 Report

The present manuscript report two news withanolides in Physalis angulata, describing the structures and cyototxic activities in cancer cell lines.

Latests publication related to Physalis angulata are not included in the manuscript (Gao et al 2018; etc.)

Table 2 are not metioned in the text. It is possible to mention it in lines 113-114.

It is necessary you mention what kind of cancer cell lines are used in the research, by other hand it would be recommendable to use a non-tumoral cell line to know a possible selective effect of compounds studied.

It is necessary to indicate the concentration of adriamycin used in the experiments of apoptosis.

Finally, some comments related to the form of the manuscript:

- In line 216 the letter type is changed

Author Response

Dear reviewer,

Thanks for your comments and suggestions. We have revised the paper according to your advices. Please see the response as follows.

This manuscript is very well written but I have some comments. First, the withanolides are known as cytotoxic agents, and the new obtained of the authors compounds do not indicate better cytotoxic effect as reference adriamycin. Secondly, the yield of the important compounds are very low and I am not sure if this compounds have perspective as drugs. 

Response:

Thank you very much for your comments on our manuscript. It is very helpful for improving our manuscript. At present we are studying some potential active compounds by the bioassay-guided methods from the genus Physalis. Withanolides showed cytotoxic activity against a variety of tumor cells.

In this paper, we’ve reported eight cytotoxic withaphysalins and physalins including two new compounds. As you mentioned, some of these compounds exhibited strong but not more potent activities compared with the reference of adriamycin. However, in some related literatures, some withanolides exhibit cytotoxic or cell-differentiation-inducing activities, and some are of great interest for the chemoprevention of cancer. Although the yields of the new compounds are low, the total steroids in the CH2Cl2 extract are not low. Withanolides are the representative constituents in the genus of Physalis. Many scientists reported the antitumor mechanisms of the cytotoxic withanolides, such as Withanolide D, Withafarin A, and Physalin B, etc. These compounds might be useful for the prevention of some types of cancers. Please check the review in “Reference [1]” that we referred in the manuscript.

Thanks!

Best wishes,

Yi Sun
